# Effects of Environmental Noise Stress on Mouse Metabolism

**DOI:** 10.3390/ijms252010985

**Published:** 2024-10-12

**Authors:** Jungmin Lee, Jehoon Yang, Jeyun Kim, Yoonjung Jang, Jisun Lee, Daehyun Han, Hunnyun Kim, Byong Chang Jeong, Je Kyung Seong

**Affiliations:** 1Laboratory of Developmental Biology and Genomics, College of Veterinary Medicine, Seoul National University, Seoul 08826, Republic of Korea; jm0511.lee@gmail.com; 2Preclinical Resource Center, Samsung Medical Center, Seoul 06351, Republic of Korea; jeyun604.kim@sbri.co.kr (J.K.); jangsuzy91@gmail.com (Y.J.); jisunny04.lee@samsung.com (J.L.); dh96.han@samsung.com (D.H.); mdpkhn.kim@samsung.com (H.K.); 3Curogen Technology, Suwon 16419, Republic of Korea; yangj@curogen.co.kr; 4College of Veterinary Medicine, Chungbuk National University, Cheong-ju 28644, Republic of Korea; 5Department of Urology, Samsung Medical Center, Sungkyunkwan University School of Medicine, Seoul 06351, Republic of Korea; 6Korea Mouse Phenotyping Center (KMPC), Seoul National University, Seoul 08826, Republic of Korea; 7Interdisciplinary Program for Bioinformatics, and BIO-MAX Institute, Seoul National University, Seoul 08826, Republic of Korea

**Keywords:** noise stress, higher locomotor activity, reduced anxiety, HDL cholesterol, adipocyte hypertrophy

## Abstract

Environmental noise is associated with various health outcomes. However, the mechanisms through which these outcomes influence behavior and metabolism remain unclear. This study investigated how environmental noise affects the liver, adipose tissue, and brain metabolic functions, leading to behavioral and body weight changes. Mice were divided into a noise group exposed to construction noise and an unexposed (control) group. Behavior and body weight changes were monitored over 50 days. Early changes in response to noise exposure were assessed by measuring plasma cortisol and glial fibrillary acidic protein expression in brain tissues on days 1, 15, and 30. Chronic responses, including changes in lipoprotein and fat metabolism and neurotransmitters, were investigated by analyzing serum lipoprotein levels and body fat mass and evaluating liver, fat, and brain tissue after 50 days. The noise group showed higher locomotor activity and reduced anxiety in the open-field and Y-maze tests. Noise exposure caused an initial weight loss; however, chronic noise increased fat mass and induced adipocyte hypertrophy. Our findings underscore the role of environmental noise-induced stress in augmenting locomotor activity and reducing anxiety in mice through neurotransmitter modulation while increasing the risk of obesity by decreasing HDL cholesterol levels and promoting adipocyte hypertrophy.

## 1. Introduction

With the rise of industrialization, environmental risk factors and past disease factors such as communicable diseases and nutritional deficiencies have become more prevalent [1]. These risk factors are increasingly being associated with various diseases. For example, air pollution has the potential to induce inflammation and oxidative stress, which can lead to cerebrovascular and neuropsychiatric disorders [2]. These emerging environmental risk factors have shifted the global disease burden from communicable to noncommunicable diseases [3]. 

Noise is a significant environmental risk factor. The WHO guidelines set the noise threshold for humans at 55 dB; damaged hearing begins at 80 dB, and noise above 130 dB can cause pain and distress [1]. Many epidemiological studies have indicated a long-term relationship between noise-induced stress and a wide range of noncommunicable diseases, such as cardiovascular [3] and metabolic diseases [4]. Moreover, noise-induced stress can lead to mental health disorders [5], independent of adverse auditory effects. Experimental data from humans strongly correlate with mechanistic data from animal models [3]. For example, mice experience auditory and nonauditory effects when exposed to noise levels above 85 dB; therefore, guidelines have been established to ensure appropriate noise control in laboratory animal facilities [6]. Noise-induced pathophysiological changes have been reported in animal models; these include increased blood pressure [7,8], increased stress hormone levels, sleep disturbances, vascular dysfunction, immune dysregulation, delayed wound healing, weight loss, and impaired fertility and reproduction [9]. These studies suggest a correlation between noise and the occurrence of noncommunicable diseases.

Noise exposure induces stress hormones, inflammatory responses, and oxidative stress responses through the sympathetic nervous system (SNS) and hypothalamic–pituitary–adrenal (HPA) axis [1]. Studies on behavioral changes caused by noise through these pathways have rarely been conducted in animals. Although behavioral changes from changes in gamma-aminobutyric acid (GABA) owing to noise stress have not been demonstrated, alterations in GABA and glutamate due to chronic noise stress have been observed in the rat brain [10]. GABA is an inhibitory neurotransmitter and a chemical messenger in the brain that has a calming effect. This hormone regulates anxiety, stress, and fear, which are associated with nerve cell hyperactivity. One of the two glutamic acid decarboxylase (GAD) proteins that contribute to the synthesis of GABA from glutamate is GAD 67 (GAD67), encoded by *GAD1* [11]. Astrocytes regulate GABA function. A crucial function of astrocytes is to regulate neurotransmitter homeostasis through the uptake of neurotransmitters such as glutamate, GABA, and glycine, released at synapses [12]. Stress can induce a primary inflammatory response [13].

Reportedly, anxiety was reduced in animal models of noise exposure due to changes in the mRNA expression of the corticotropin-releasing hormone (CRH) system, a stress hormone in the brain; however, the mechanism remains unclear [14]. Acute stress increases cortisol levels [15]. Changes in glucocorticoid biomarkers and neurotransmitters can accompany symptoms of stress-induced anxiety disorder due to activation of the SNS and the HPA axis [16]. 

Chronic stress has been associated with lipoprotein and fat metabolism [17,18]. Scavenger receptor class B type 1 (SR-B1) is a high-density lipoprotein (HDL) receptor and multiligand membrane receptor protein expressed in the adrenal gland, liver, and adipose tissue. The primary role of this receptor is to selectively deliver HDL-derived cholesteryl esters and other lipids, such as free cholesterol and triglyceride molecules, from HDL to cells and tissues [19]. Hepatic overexpression of SR-B1 in mice results in the disappearance of plasma HDL cholesterol [20]. The expression of SR-B1 increases during stress response, facilitating cholesterol uptake via HDL clearance and utilizing it for glucocorticoid synthesis [21]. 

Leptin, derived from adipose tissue, acts as an anti-steatosis hormone by inhibiting lipid accumulation in the liver and promoting its elimination [22]. Leptin levels increase in obesity. Additionally, although acute stress decreases leptin levels [23], chronic stress increases it [24]. Serum leptin and corticosterone levels have been reported to increase in noise-exposed rats [25], and, epidemiologically, serum leptin levels reportedly increase during mental stress [26]. Leptin is also an important regulator of hepatic SR-B1 expression and HDL cholesterol levels, and plasma cholesterol levels increase in leptin-deficient ob/ob mice treated with leptin [27].

Although stress hormones have been reported to cause behavioral and body weight changes, it is important to confirm whether noise causes acute changes in cortisol levels, resulting in a stress response. Additionally, it is essential to investigate the potential effects of early-stage noise exposure on brain homeostasis, which may result in chronic neurotransmitter alterations and consequent behavioral changes. Furthermore, it is imperative to determine the chronic effect of noise-induced stress on fat and lipoprotein metabolism concerning body weight changes during the period of noise exposure and to observe the relationship between changes in adipose tissue and blood lipoproteins and SR-B1 and leptin expression in the liver.

In this study, we investigated the effects of environmental noise on metabolic functions in the liver, adipose tissue, and brain, which are associated with behavioral and body weight changes in mice exposed early to chronic noise.

## 2. Results

### 2.1. Noise Exposure Increases Locomotor Activity and Reduces Anxiety

We assessed locomotor activity in an open field in the noise and control group once a week for 5 weeks (days 7, 14, 21, 28, and 35) during the noise exposure period (Figure 1A). In the open field test, there was no significant difference in the total travel distance between the two groups until the evaluation on day 28 of noise exposure (Control = 1986.97 ± 760.87 cm vs. Noise = 2126.92 ± 818.43 cm, *p* = 0.631; Figure 1B). In the repeated weekly evaluations, both groups showed a tendency to decrease the distance traveled until day 28. However, it was confirmed that the distance moved significantly increased in the noise group on day 35 of noise exposure (Control = 1771.61 ± 714.90 cm vs. Noise = 2507.11 ± 804.49 cm, *p* = 0.013; Figure 1B). On day 35, the distance traveled by the noise group increased to a similar extent as that of day 14, deviating from the decreasing trend. In the control group, following a decreasing trend, the distance traveled on day 35 was the lowest compared to days 7, 14, 21, and 28.

After analyzing the total travel distance divided into the periphery and center, a similar trend was observed in the total travel distance graph. The travel distance of the noise group increased in the periphery (Control = 1696.40 ± 653.80 cm vs. Noise = 2315.63 ± 665.65 cm, *p* = 0.027; Figure 1C) and the center on day 35 (Control = 75.22 ± 84.35 cm vs. Noise = 191.48 ± 217.96 cm, *p* = 0.006; Figure 1D).

The control group showed decreased movement traces in the open-field arena between days 7 and 35 of noise exposure compared to that in the noise group. However, the noise group had more movement traces than the control group on day 35 (Figure 1E). 

Regarding rearing behavior, the control group showed a tendency for weekly decrease, similar to the trend in the distance traveled. However, in the noise group, the distance decreased until day 21 and increased from day 28 onward. The number of rears on days 7, 14, and 35 was significantly higher in the noise group than that in the control group (172.93 ± 65.31 vs. 127.20 ± 55.93; *p* = 0.049, 142.60 ± 54.82 vs. 77.93 ± 33.13; *p* < 0.001, and 92.80 ± 37.24 vs. 58.27 ± 42.05; *p* = 0.024, respectively; Figure 1F).

Regarding time spent at the center, both groups tended to decrease weekly measurements until day 28. However, on day 35, the time spent in the center increased in the noise group, similar to that measured on day 14. Hence, on day 35, the noise group stayed at the center significantly longer than the control group (48.71 ± 72.04 s vs. 8.71 ± 11.52 s, *p* = 0.043; Figure 1G).

In the Y-maze test, the latency time taken to exit the starting arm was significantly reduced in the noise group compared to that in the control group (9.62 ± 16.45 s vs. 132.76 ± 135.52 s, *p* = 0.002; Figure 1H).

However, no differences were observed between the two groups regarding spontaneous alternation performance in the Y-maze (Control = 67.16% ± 20.40% vs. Noise = 73.78% ± 8.91%, *p* = 0.259; Figure 1I).

### 2.2. Noise Exposure Reduces Body Weight 

The two groups had no difference in body weight after random group assignment. However, on days 2, 23, and 28 of noise exposure, the average weight gain in the noise group was significantly lower than that in the normal group (Figure 2A). From day 29 onward, the average weight of the noise group was lower; however, there was no significant difference in weight between the two groups.

### 2.3. Noise Exposure Increases Plasma Cortisol Levels

Plasma cortisol levels were measured on days 1, 15, and 30 of noise exposure to assess the effects of noise exposure on plasma cortisol levels in male and female mice. Plasma cortisol levels increased the most on day 1 and decreased from days 15 to 30 as the noise exposure period was extended in the noise group for both males and females. In the male mice, cortisol levels significantly increased in the noise group on day 1 of noise exposure compared to that in the control group (77.76 ± 39.40 ng/mL vs. 14.39 ± 16.68 ng/mL, *p* = 0.024; Figure 2B). Similarly, in the female mice, the plasma cortisol levels were significantly higher in the noise than in the control group on day 15 after noise exposure (30.04 ± 15.34 ng/mL vs. 7.65 ± 2.63 ng/mL, *p* = 0.028; Figure 2C).

### 2.4. Noise Exposure Significantly Reduced the Serum Levels of HDL Cholesterol and Triglyceride and Promoted the Expression of SR-B1 and Leptin in the Liver

Serum levels of HDL cholesterol (Control = 47.43 ± 8.02 mg/dL vs. Noise = 39.79 ± 4.85 mg/dL, *p* = 0.006; Figure 2D) and triglyceride (Control = 147.36 ± 45.45 mg/dL vs. Noise = 97.93 ± 29.56 mg/dL, *p* = 0.004; Figure 2E) significantly decreased in the noise group; however, there was no difference in the serum levels of total cholesterol (Control = 76.07 ± 7.73 mg/dL vs. Noise = 71.93 ± 5.24 mg/dL, *p* = 0.115; Figure 2F) and glucose (Control = 219.07 ± 22.18 mg/dL vs. Noise = 210.07 ± 31.48 mg/dL, *p* = 0.468; Figure 2G) between the two groups. After prolonged noise exposure, SR-B1 (% pixels of SR-B1/Total pixels, Control = 19.72% ± 1.84% vs. Noise = 21.60% ± 2.20%, *p* < 0.001; Figure 2H,I) and leptin (% pixels of leptin/total pixels, Control = 1.37% ± 1.48% vs. Noise = 4.90% ± 4.23%, *p* < 0.001; Figure 2J,K) expression in the liver was significantly higher in the noise group than that in the control group. 

### 2.5. Noise Exposure Positively Correlates with Increased Fat Mass and Adipocyte Hypertrophy

Measurement of body composition using dual-energy X-ray absorptiometry (DEXA) demonstrated an increase in fat mass in the noise group compared to that in the controls (5.20 ± 0.41 g vs. 4.72 ± 0.38 g, *p* = 0.002, Figure 3A). Notably, lean body mass remained similar in both groups. The fat percentage significantly increased in the noise group compared to the control group (21.91% ± 2.10%, vs. 19.51% ± 1.67%, *p* = 0.002; Figure 3B).

Average adipocyte size significantly increased in the noise group compared to that in the control group (838.49 ± 142.86 μm^2^ vs. 703.11 ± 157.62 μm^2^, *p* = 0.015; Figure 3C). We found that large-sized adipocytes were significantly more distributed in the noise group than they were in the control group (adipocyte area, 0–500 μm^2^, Control = 48.15% ± 10.31% vs. Noise = 44.74% ± 9.30%, mean ± standard deviation (SD), *p* = 0.044; adipocyte area 500–1000 μm^2^, Control = 28.10% ± 5.38% vs. Noise = 26.00% ± 4.10%, mean ± SD, *p* = 0.011; adipocyte area 1500–2000 μm^2^, Control = 5.48% ± 3.92% vs. Noise = 7.70% ± 3.81%, mean ± SD, *p* = 0.001; adipocyte area 2000–3000 μm^2^, Control = 3.52% ± 3.42% vs. Noise = 4.97% ± 3.86%, mean ± SD, *p* = 0.021; Figure 3D,E).

### 2.6. Noise Exposure Decreases GAD 67 Expression in Mice Brain

To assess the effects of noise exposure on tyrosine hydroxylase (TH) and GAD67 expression in the mouse brain, immunohistochemical analyses of TH and GAD67 expression were conducted. Most cells were TH-positive in the substantia nigra pars compacta (SNc) and ventral tegmental area (VTA) and GAD67-positive in the substantia nigra pars reticulata (SNr) (Figure 4A,B). The VTA, SNc, and SNr were identified at the bregma −2.80 level.

The total area stained with anti-TH in the coronal section at the level of bregma −2.80 level was compared between groups. The TH-positive staining area in the noise group was more increased than that in the control group, but no significant differences were found between the groups (*t*-test, % of pixels of TH/Total pixels, Control = 0.61% ± 0.21% vs. Noise = 0.77% ± 0.39%, mean ± SD, *p* = 0.168; Figure 4C).

Similarly, the total area stained with anti-GAD67 in the coronal section at the bregma −2.80 level was compared between groups. The GAD67-positive stained area in the noise group was significantly decreased compared to that in the control group (5.20% ± 1.95% vs. 6.81% ± 2.15%, *p* = 0.043; Figure 4D).

To analyze the effect of noise exposure on glial fibrillary acidic protein (GFAP) expression in the mouse brain, immunohistochemical analysis of GFAP expression was performed on days 1, 15, and 30 after noise exposure. GFAP was more expressed in the noise group in the SNr (Figure 4E).

## 3. Discussion

Pathophysiological changes caused by noise exposure activate the SNS and HPA axis [1]. This study aimed to investigate the early and chronic [10,28] effects of noise exposure in animals to clarify the stress reactions via the SNS and HPA axis. The present study exposed mice to noise from a construction environment for 50 days. We observed higher locomotor activity, reduced anxiety caused by the downregulation of inhibitory neurotransmitters, and increased risk of obesity due to decreased HDL cholesterol and the promotion of adipocyte hypertrophy. 

Detecting stress-related anxiety in rodents is based on species-specific behaviors such as reduced exploration, shelter-seeking, escape, burying, or defecating [29]. The noise group showed hyperactive and anxiolytic movements in the behavioral tests. The mice in this group also showed more spontaneous motor activity, spent more time in the center, showed increased rearing in the open-field test, and had shorter latencies to exit the starting arm in the Y-maze test than those in the control group.

An increase in cortisol levels, an important stress biomarker, was observed on day 1 after noise exposure, whereas most behavioral alterations were observed on day 35 after noise exposure. However, cortisol levels appeared to return to normal 30 days after noise exposure. Plasma cortisol was the highest on day 1 in the noise group for both male and female mice and tended to decrease on days 15 and 30. Comparing the control group, cortisol levels significantly increased in the noise group on days 1 and 15 of noise exposure in male and female mice, respectively. After 30 days of noise exposure, the plasma cortisol levels of both males and females in the noise group were found to be equivalent to those of the control group. High plasma cortisol levels are important biomarkers of excessive stress. It has been reported that 30 min of acute noise exposure in rats increases the levels of corticosterone, an adrenocorticotropic hormone [30]. However, epidemiological studies have not reported any impact of cortisol levels on the health effects of noise exposure [31]. In the present study, cortisol levels increased upon initial exposure and decreased as the animals became accustomed to the environmental noise; this suggests that the rise in cortisol levels due to noise exposure occurs acutely and subsequently decreases to the level observed in the control group over time. Therefore, it does not appear to be directly related to the behavioral changes or increased risk of obesity observed in this study. Given these results, cortisol concentrations may be a feasible marker for acute stress, but their use as a marker for chronic stress remains controversial. Interestingly, the different timing of peak cortisol levels between males and females suggests that the stress response may vary by sex.

After noise exposure, GFAP expression increased in the brain on days 1, 15, and 30. GFAP is a marker for astrocytes that removes glutamate from synapses. Glutamate is the main excitatory neurotransmitter and the metabolic precursor of GABA, the predominant inhibitory neurotransmitter [32]. Noise stress may reduce GABA production by decreasing glutamate levels. However, quantitative analysis of GFAP expression increases was not possible in this study. Although GFAP expression appeared higher on days 1 or 15 compared to day 30, further studies are needed to identify early changes in astrocyte activation because of noise stress.

Dopaminergic neurons are mainly distributed in the SNc and VTA [33]. Brain tissue sections were immunostained and observed at −2.80 mm relative to bregma, where the SNc and VTA could be identified. The expression of TH, which represents dopaminergic neurons, increased in the SNc and VTA. Glutamatergic and GABAergic neurons are predominantly distributed in the SNr [34], and the expression of GAD67, which signifies both glutamatergic and GABAergic neurons, was significantly reduced in the SNr. Consistent with our findings in this study, noise exposure has been reported to upregulate TH, an enzyme responsible for dopamine synthesis [20], reduce GABA levels, and upregulate glutamic acid levels in the hippocampus [21]. 

Dopamine has both excitatory and inhibitory effects, whereas GABA has only a major inhibitory effect on the brain. Neuronal hyperactivity associated with anxiety, stress, and fear is regulated by GABA, which has a calming effect. GAD67 enzyme converts glutamate to GABA. After 1 week of restraint stress in mice lacking dopamine D4 receptors (2 h per day), schizophrenia phenotypes such as elevated exploratory behaviors were observed, and GABAergic transmission was significantly reduced [35]. Mice with a disrupted Gad67 allele display mild hyperactivity [11]. Brain GABA and GAD67 levels are reportedly reduced in chronically stressed animals [36]. Our results confirm that noise-induced stress initially causes an increase in plasma cortisol, activates astrocytes, and chronically induces changes in the balance of inhibitory neurotransmission, resulting in behavioral changes; this reduces the conversion of glutamate, an excitatory neurotransmitter, to GABA, an inhibitory neurotransmitter, by GAD67, thereby inducing a state of reduced inhibitory neurotransmitters, which results in higher locomotor activity and reduced anxiety. In this study, we found that GAD67 was significantly decreased by noise stress, likely because of a reduction in GABA, which is consistent with the behavioral assessment results of increased activity and decreased anxiety. This demonstrates that neurotransmitter changes caused by the stress of real-world noise exposure lead to corresponding behavioral changes.

Common responses to acute stress include reduced food intake, increased heat production, and hyperactivity; however, when chronic stress progresses, it may cause weight gain in restrained eaters with increased HPA reactivity [37]. Weight loss due to noise exposure for 19.5 days has also been reported in rats [38]. Recent epidemiological studies have linked chronic noise stress to obesity, high cholesterol levels, and diabetes [28,39,40]. Consistent with these findings, our study showed reduced weight gain in the early stages of noise exposure; however, signs of obesity appeared as noise exposure progressed. Increased fat mass and adipocyte hypertrophy were confirmed in the noise group. In this study, on day 50 of noise exposure, leptin expression was significantly increased in the liver. Considering that the typical phenomena leading to obesity are increased fat mass, adipocyte hypertrophy, and a decreased response to leptin [41], it can be hypothesized that obesity was initiated by chronic noise stress.

The stress response increases the expression of SR-B1, which supplies cholesterol through HDL clearance and is used to produce glucocorticoids [21]. In the present study, chronic noise stress increased the expression of SR-B1 in the liver and decreased serum HDL cholesterol levels. There is a link between obesity and cholesterol absorption, synthesis, lipoprotein processing, and hepatic cholesterol accumulation [42]. Noise stress causes chronic fat and lipoprotein metabolism changes, leading to obesity.

Our results strongly suggest that noise-induced stress causes changes in early plasma cortisol levels and astrocytes, impairing chronic inhibitory neurotransmitter activity, thereby increasing locomotor activity and reducing anxiety. Although there are clinical and epidemiological reports related to noise stress, limited research has been conducted in this field. Moreover, it is debatable whether noise-induced stress consistently increases plasma cortisol [3] and causes reduced physical activity in humans [43]. Therefore, additional studies are necessary to elucidate the mechanisms underlying behavioral changes based on the duration of noise stress exposure. For an objective comparison of neurotransmitter expression in the mouse brain, we measured the expression of TH and GAD67 at the same location, at the bregma −2.80 level, and compared these between groups. However, our study did not examine the expression of TH or GAD67 across the entire mouse brain. Given the significant differences in GAD67 expression between groups, further research is warranted on noise-induced changes in neurotransmitter expression, using methods such as 3D whole-brain imaging approaches [44]. Furthermore, it is imperative to acknowledge and analyze the potential disparities in results that may arise due to differences in noise perception between human and animal experiments. It is necessary to identify the mechanisms associated with inflammation and oxidative stress, in which initial weight loss due to noise stress leads to chronic obesity. Additionally, noise stress reportedly affects the immune system in humans [31]; therefore, further research is warranted to examine the effects of noise stress on the immune system.

## 4. Materials and Methods

### 4.1. Animals 

The mice were randomly assigned to the noise and control groups with no difference in body weight. Eight mice per group (four males and four females) were euthanized on days 1, 15, and 30 to determine plasma cortisol concentrations and observe early changes in response to noise exposure. In male mice, brains from 2 to 4 mice per group were sampled on days 1, 15, and 30 post-noise exposure to examine GFAP expression. Fifteen male mice per group were assessed for noise-induced chronic changes. Behavioral and body weight assessments were performed serially during the noise exposure period. For the evaluation of chronic changes in liver and adipose tissue, as well as TH and GAD67 expression in the brain, 15 male mice per group were euthanized, and tissue samples were collected. All mice used in this study to observe early and chronic changes were 5-week-old C57BL/6N mice, purchased from Orient Bio Co., Ltd. (Seongnam, Korea). After a 1-week acclimatization and stabilization period, the animals were used for experiments according to the animal facility’s standard operating procedures.

Considering unexpected abnormalities that may occur with prolonged noise exposure and variations in behavioral evaluation results, the number of animals in the group being assessed for gradual and chronic effects was designed to include approximately twice as many animals as that of the group observed for early effects. The Institutional Animal Care and Use Committee of Samsung Medical Center reviewed and approved this study protocol (approval number: 20191126001). 

### 4.2. Experimental Design

All animals were housed under specific pathogen-free (SPF) conditions and provided a regular chow diet (PicoLab^®^ Rodent Diet 20, LabDiet, St. Louis, MO, USA) and reverse osmosis water ad libitum. The animal room was maintained at 20–22 °C, with 40–60% relative humidity, an air exchange rate of 12–18 per h, and a 12:12 h light:dark cycle. The mice were housed in individually ventilated cages with a noise level of <60 dB. 

The experiments for the noise-exposed and control groups were conducted simultaneously. Both groups were housed in animal rooms with a background noise level of 60 dB or less. However, the noise-exposed group was moved near the remodeling area in the vivarium that generated 70 to 100 dB of noise from 8 a.m. to 12 p.m. Monday through Friday and was intermittently exposed to construction noise of 70–100 dB for 50 days. After noise exposure, the mice were kept in the same animal room as the control group. 

Body weight was measured thrice weekly for the 50 days of noise exposure. The open field test was conducted on days 7, 14, 28, and 35, and the Y-maze test was conducted on day 20. All mice were acclimated to the experimental environment for 30 min before behavioral tests. 

To examine chronic changes after noise exposure, the DEXA test, serum chemistry, and histopathological analyses were performed on days 49 and 50. 

### 4.3. Cortisol Measurement

Cardiac blood sampling in male and female mice was conducted under deep terminal anesthesia on days 1, 15, and 30 after the noise exposure. Blood was collected in heparin-coated tubes and centrifuged at 3400 rcf for 15 min to collect the plasma. The collected plasma samples were stored at −80 °C for later use. A mouse cortisol ELISA kit (MBS269130; MyBioSource, Inc., San Diego, CA, USA) was used to measure the plasma cortisol levels according to the manufacturer’s protocol.

### 4.4. Open Field Test

The locomotor activity and anxiety of the mice were assessed using an Auto-Track Opto-Varimex activity monitoring system (Columbus Instruments, Columbus, OH, USA). The mice were allowed to explore an open-field arena (44.5 cm × 44.5 cm) for 20 min. The analysis was performed using Auto-Track software (Version 5.00, Columbus Instruments, Columbus, OH, USA), where the arena was divided into a center (28.5 × 28.5 cm) of an area in the central zone of the arena and a periphery excluding the center. Behavioral patterns in the open field test were evaluated by measuring traces of movement, exploration distance, retention time in the center, and the number of rears. After the noise exposure, the test was performed on days 7, 14, 21, 28, and 35.

### 4.5. Y-Maze Test

The anxiety and spatial memory of the mice were assessed using the Y-maze test on day 20 of noise exposure. A Y-shaped maze was used with three white opaque plastic arms (35 × 5 × 10 cm) attached at an angle of 120°. An automated tracking system (EthoVision XT 10, Noldus, Wageningen, The Netherlands) was used to monitor the movement of the mice for 8 min. The mouse was placed at the entrance to one of the three arms facing away from the center and allowed to move through the apparatus. Behavioral patterns in the Y-maze test were evaluated by measuring the latency time to exit the starting arm and the number of triads to calculate the percentage of alternation. 

### 4.6. DEXA 

DEXA imaging was conducted to determine the body composition of the mice on day 49 after noise exposure. The DEXA equipment (Lunar PIXImus2, GE Lunar Corp., Madison, WI, USA) was calibrated using a plastic phantom mouse for quality control provided by the manufacturer, and the main imaging was performed after the equipment passed the quality control. Mice were anesthetized intraperitoneally using a combination of zolazepam and tiletamine as Zoletil 50^®^ (25 mg/kg, Virbac, Carros, France) and xylazine as Rompun^®^ (10 mg/kg, Bayer, Leverkusen, Germany). The animals, including their tails, were placed dorso-ventrally on a DEXA tray. At this point, the head was slightly outside the field of view. X-ray irradiation was performed four times at low (40 kV) and high (70 kV) energy, respectively, and averaged. The percentage of body fat and composition, excluding the head, were calculated automatically.

### 4.7. Serum Chemistry 

To determine whether noise exposure affects the function of each organ, blood was collected at the time of autopsy on day 50 and separated into serum. Serum glucose, total cholesterol, HDL cholesterol, and triglyceride levels were measured using a DRI-CHEM 7000i analyzer (Fujifilm, Tokyo, Japan).

### 4.8. Histology and Immunohistochemistry

Anesthetized mice were transcardially perfused with saline, and brain, liver, and inguinal adipose tissues were harvested and fixed in 10% neutral buffered formalin. Adipocyte measurement was performed by slicing paraffin-embedded fat tissues into 4 µm sections, deparaffinized in xylene, rehydrated in graded alcohol series, and stained with hematoxylin and eosin. 

Brain and liver tissues were sliced into 4 µm sections for immunohistochemical analysis, processed as described above, and transferred to 0.01 M phosphate-buffered saline (PBS, pH 7.4). Using citrate buffer (pH 6.0; Dako, Carpinteria, CA, USA) at 121 °C for 3 min, heat-induced epitope retrieval (HIER) was performed to reveal hidden antigen epitopes. After washing with PBS, 3% hydrogen peroxide in PBS was added at 20–24 °C for 10 min to block endogenous peroxidase activity. To block nonspecific binding, sections were washed with PBS and treated with a serum-free blocking solution (Dako) at RT for 30 min. 

Immunohistochemical localization of the GFAP, TH, or GAD67 was performed by incubating coronally sliced brain sections with rabbit polyclonal anti-GFAP antibody (diluted 1:3000; Abcam, Cambridge, UK), mouse monoclonal anti-TH antibody (diluted 1:100; Millipore, Burlington, MA, USA), and mouse monoclonal anti-GAD67 antibody (diluted 1:1000: Merck, Rahway, NJ, USA) at 4 °C overnight. The immunohistochemical localization of leptin or SR-B1 was performed by incubating liver sections with rabbit polyclonal anti-leptin (diluted 1:300; Bioss, bs-0108R, Boston, MA, USA) and anti-SR-B1 (diluted 1:1000; Abcam, Cambridge, UK) antibodies at 4 °C overnight. After washing with PBS, all the sections were incubated with an HRP-labeled polymer conjugated to anti-mouse IgG (for TH and GAD67; Dako, Wiesentheid, Germany) or anti-rabbit IgG (for GFAP, leptin, and SR-B1; Dako) at RT for 30 min. Color reactions were developed using the ready-to-use 3,3′-diaminobenzidine (DAB) substrate-chromogen solution (Dako) for 3 min, and the sections were washed with distilled water. Finally, the sections were lightly counterstained with Mayer’s hematoxylin (Dako) for 30 s before dehydration and mounting. Tissue sections were analyzed under a microscope (Olympus Co., Tokyo, Japan) with a digital camera (DP73, Olympus). 

TH and GAD67 expression in control and noise-exposed mice was analyzed by measuring fields from each section using the ImageJ software (Version 1.53e, National Institutes of Health, Bethesda, MD, USA). Images of coronal sections at the bregma −2.80 level stained for TH and GAD67 were obtained for each individual and compared between groups. For each image, the intensity was subtracted from the background. The percentage of areas with positive staining intensity was calculated.

Adipocyte size was quantified by selecting five slides from images for each subject using ImageJ. The number of adipocytes on each slide and the area data for each adipocyte were obtained using ImageJ [45]. To analyze leptin and SR-B1 expression, three slides were selected for each subject and analyzed using ImageJ.

### 4.9. Statistical Analysis

All results are expressed as the mean ± SD. Data analysis and two-group comparisons were performed using Student’s *t*-test in Microsoft Excel 2016 (Microsoft Corporation, Redmond, WA, USA) software. Statistical significance was set at *p* < 0.05.

## Figures and Tables

**Figure 1 ijms-25-10985-f001:**
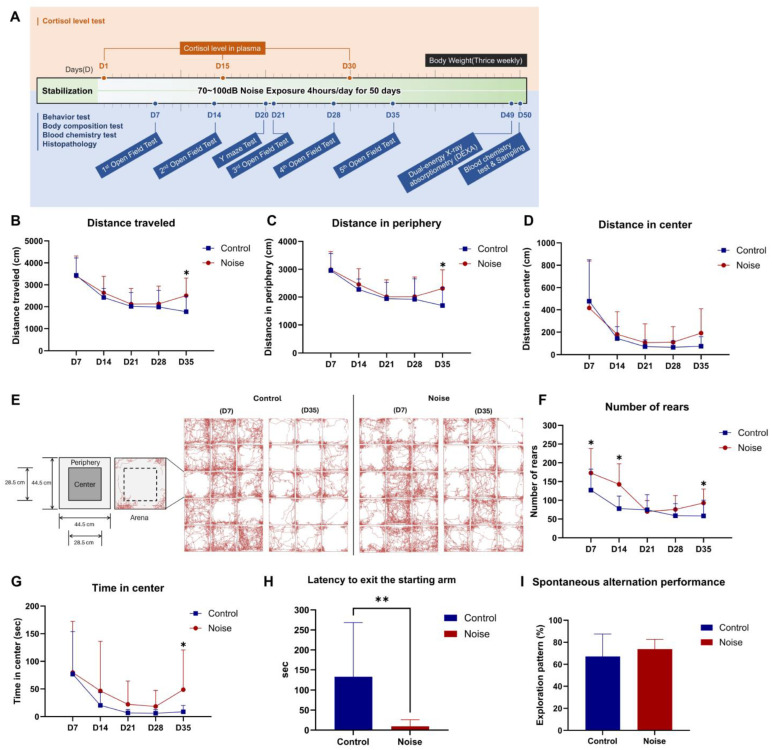
The noise group exhibited higher locomotor activities and anxiolytic behaviors. (**A**) The experimental scheme for noise exposure. (**B**) Changes in total movement distance during the Open Field test. The total movement distance of the noise group significantly increased on day 35 (D35). (**C**) Changes of movement distance in the peripheral zone in the Open Field test. The movement distance of the noise group in the peripheral zone was significantly longer than that of the control group on day 35 (D35). (**D**) Changes of movement distance in the central zone in the Open Field test. (**E**) Traces of spontaneous movements in the Open Field arena. Each trace represents the total distance traveled by the subject during the 20 min period of the test. In the control group, locomotor activity decreased on day 35 (D35) compared to day 7 (D7). Conversely, the noise group showed less locomotor activity reduction than the control group during the same period. (**F**) Changes of rearing behavior in the Open Field test. The number of rears in the noise group significantly increased on days 7 (D7), 14 (D14), and 35 (D35). (**G**) Changes in time spent within the central zone in the Open Field test. The noise group stayed significantly longer in the central zone on day 35 (D35) than the control group. (**H**) The latency time to exit the starting arm was significantly shorter in the noise group during the Y-maze test. (**I**) Spontaneous alternation performance in the Y-maze test showed no significant difference; this means there is no difference in spatial memory between the control and the noise groups (*n* = 15 per group). Data are presented as the mean ± standard deviation (SD). Student’s *t*-test was used to compare the means from the two groups (control group vs. noise group under the same food and environmental temperature conditions). * *p* < 0.05, ** *p* < 0.01 indicate significant differences compared to the control group mice.

**Figure 2 ijms-25-10985-f002:**
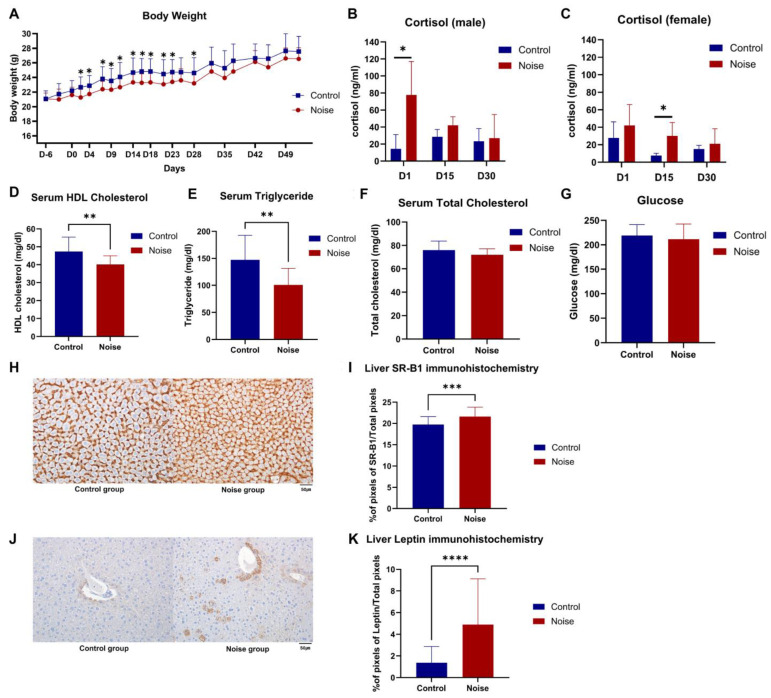
Effect of chronic noise exposure on body weight, plasma cortisol level, serum high-density lipoprotein (HDL) cholesterol, and expression of scavenger receptor class B type 1 (SR-B1) and leptin in the liver. (**A**) Changes in body weight during noise exposure. Reduced body weight was observed in the noise group from day 2 (D2) to 23 (D23) and 28 (D28). (**B**) In male mice, cortisol levels significantly increased on day 1 after noise exposure (D1). (**C**) In female mice, cortisol levels significantly increased on day 15 after noise exposure (D15). (**D**) The HDL cholesterol was significantly decreased in the noise group. (**E**) Triglyceride levels were significantly decreased in the noise group. (**F**) The serum total cholesterol level remained similar in the noise and control groups. (**G**) Blood glucose level on day 50 (the end of the noise exposure period). Noise exposure did not affect blood glucose levels. (**H**) SR-B1 staining in the noise group was stronger than that in the control group. (**I**) The stained SR-B1 area to total area measured with ImageJ was markedly increased in the noise group. (**J**) Leptin staining in the noise group was stronger than that in the control group. (**K**) The stained leptin area to total area measured with ImageJ was markedly increased in the noise group (*n* = 12–15 per group). Data are presented as the mean ± standard deviation (SD). Student’s *t*-test was used to compare the means from the two groups (control group vs. noise group under the same food and environmental temperature conditions). * *p* < 0.05, ** *p* < 0.01, *** *p* < 0.001, **** *p* < 0.0001 indicate significant differences compared to the control group mice.

**Figure 3 ijms-25-10985-f003:**
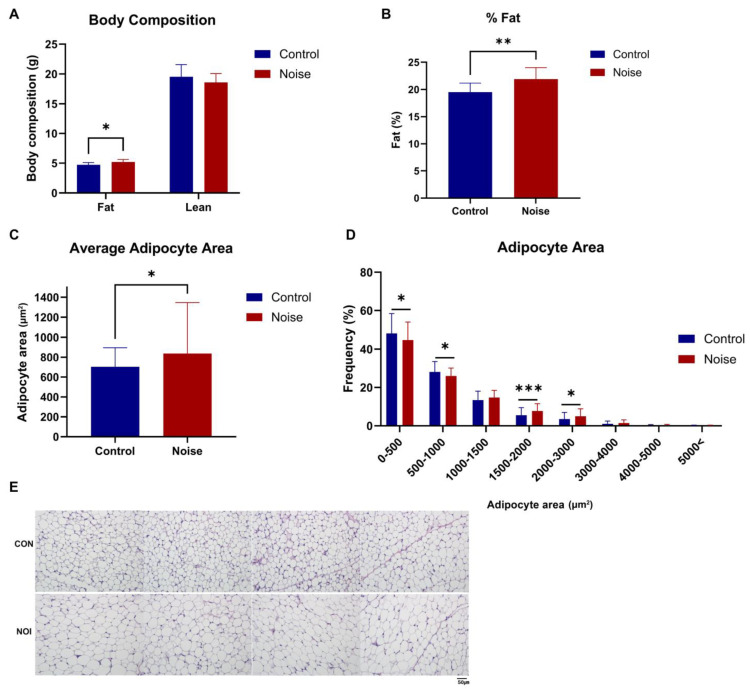
Chronic noise exposure positively correlated with increased total fat mass and adipocyte hypertrophy. (**A**) Although the total lean mass remained similar in both groups, the total fat mass significantly increased in the noise group. (**B**) The percentage of body fat also increased in the noise group. (**C**) The average adipocyte area measured with ImageJ significantly increased in the noise group. (**D**) There was a lesser frequency of small adipocytes (<1000 μm^2^) and a higher frequency of large adipocytes (1500–3000 μm^2^) in the noise group. (**E**) The difference in adipocyte size is evident in the representative H&E-stained inguinal adipose tissue (n = 15 per group). Data are presented as the mean ± standard deviation (SD). Student’s *t*-test was used to compare the means from two groups (control group vs. noise group under the same food and environmental temperature conditions). * *p* < 0.05, ** *p* < 0.01, *** *p* < 0.001 indicate significant differences compared to the control group mice.

**Figure 4 ijms-25-10985-f004:**
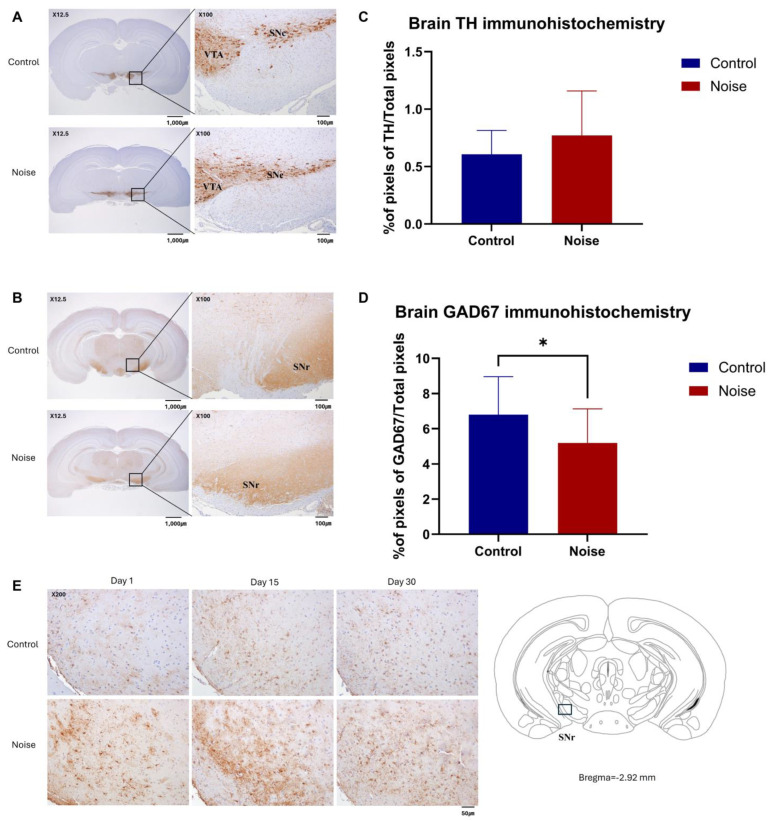
Immunohistochemical analysis of tyrosine hydroxylase (TH), glutamic acid decarboxylase67 (GAD67), and glial fibrillary acidic protein (GFAP) expression in control and noise-exposed mice. (**A**) Images of anti-TH staining in the substantia nigra pars compacta (SNc) and ventral tegmental area (VTA). (**B**) Images of anti-GAD67 staining in substantia nigra pars reticulata (SNr). (**C**) The stained TH area to the total area measured with ImageJ increased in the noise group. However, no significant differences were found between the two groups. (**D**) The stained GAD67 area to the total area measured with ImageJ was decreased in the noise group (*n* = 15 per group). (**E**) The stained GFAP area increased in the noise group. Data are presented as the mean ± standard deviation (SD). Student’s *t*-test was used to compare the means from the two groups (control group vs. noise group under the same food and environmental temperature conditions). * indicates significant differences (*p* < 0.05) compared to the control group mice.

## Data Availability

The datasets used and analyzed during the current study are available from the corresponding author on reasonable request.

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
