# Peer review of "Effects of Environmental Noise Stress on Mouse Metabolism"

_ijms, 2024, doi:10.3390/ijms252010985_

Round 1

Reviewer 1 Report

Comments and Suggestions for Authors

I appreciate the authors' work and find the results intriguing; however, I must point out several concerns that warrant attention in this review.

Comment 1: Experimental Design Clarity

The article lacks clarity regarding the experimental design, specifically whether the noise and non-noise experimental groups were studied in parallel. It is essential to confirm whether both groups were assessed simultaneously or if the noise group was evaluated before or after the non-noise group. This ambiguity could significantly impact the interpretation of the results and should be addressed to enhance the study's rigor.

Comment 2: Figure Legends and Presentation

In Figure 1, there is an absence of a comprehensive legend for the line graphs. Currently, the legend pertains only to one line graph, which is not standard practice. Typically, a legend should be included at the beginning of the figure with a statement indicating that it applies to all relevant graphs (e.g., "This legend applies to Figures XX to YY"). It is noteworthy that Figure 2 adheres to proper legend formatting, making this inconsistency in Figure 1 particularly puzzling. This oversight should be rectified for clarity and consistency.

Comment 3: Methodology for Stain Intensity Measurement

The method described by the authors for measuring stain intensity appears to be highly subjective and potentially non-reproducible. The authors state that “Images with the highest TH expression in the SNc and VTA regions were selected for all mice.” This selection process raises concerns about bias. To improve reproducibility and objectivity, it is recommended that the authors quantify expression levels across all slides using stereological methods, which provide a more precise measurement. Alternatively, conducting whole-brain sectioning and measuring the volume of stained areas would yield more reliable data.

Comment 4: Selection of Staining Areas

The rationale behind selecting the area of most intense staining in one hemisphere is unclear. This approach suggests a preference for the most confident part of the brain, which may introduce bias into the results. Modern techniques allow for serial or thick sectioning of brain tissue, enabling visualization of staining levels throughout the entire brain. The authors should consider employing these methods to provide a more comprehensive analysis that respects the complexity of brain structure. Additionally, as suggested by Paweł Kaczmarek (https://orcid.org/0000-0003-0419-6330), whole-organ 3D reconstruction methodologies or light-sheet imaging techniques could be beneficial alternatives.

Comment 5: Sample Size and Organ Analysis

There is also a lack of information regarding how many organs were analyzed in this study. It remains unclear whether the authors sectioned and analyzed brains, livers, or adipose tissues from one, ten, or twenty mice. Providing this information is crucial for understanding the scope and reliability of the findings.

Comment 6: Measurement Parameters in ImageJ

In the Materials and Methods section, the authors mention using ImageJ to measure adipocyte tissue but fail to specify what parameters were measured (e.g., length, area). It is crucial to clarify which axes were analyzed and how these measurements contribute to understanding adipocyte characteristics. Providing this information will enhance transparency and reproducibility in their methodology.

Conclusion

Overall, while the article presents interesting findings, several methodological concerns need to be addressed to improve clarity, reproducibility, and scientific rigor. While I find that behavioral activity is well-written with no concerns, I must emphasize that image analysis and sample selection represent the weakest aspects of this paper and require improvement.

Reviewer 2 Report

Comments and Suggestions for Authors

Jungmin Lee et al. investigated how environmental noise affects the liver, adipose tissue, and brain metabolic functions, leading to behavioral and body weight changes.I think this is a good manuscript, but it needs to address the following issues before it is published.

1.All bar charts or line charts in the manuscript were used in black and the authors can modify them to color.

2.The authors need to improve the Figure 1A because it is not suitable and aesthetic.

3.In this study, the authors selected 5 weeks of old mice. Are the mice too young?.

4.The authors need to add a graphical abstract.

5.The authors should revise the discussion section, the discussion section seems to be repeating the experimental results.

Reviewer 3 Report

Comments and Suggestions for Authors

The purpose of this study was to investigate the ways in which environmental noise influences the metabolic activities of the brain, liver, and adipose tissue, which in turn leads to changes in behavior and body weight. The study is well-designed, and the findings are presented effectively. I completely support the background of this work and understand its importance and relevance.

1)     Please describe the process used to set up the noise exposure.

2)     Did the authors assess the effects of prolonged noise exposure on mice's glucose, lipid metabolism and serum metabolites?

Round 2

Reviewer 1 Report

Comments and Suggestions for Authors

Thank you for addressing my concerns. I believe that your paper is now suitable for publication.

Great work!

Best regards,